# *Pseudognaphalium affine* Extract Alleviates COPD by Inhibiting the Inflammatory Response via Downregulation of NF-κB

**DOI:** 10.3390/molecules27238243

**Published:** 2022-11-26

**Authors:** Xiangli Ye, Shuping Luo, Xiaona Chang, Yaling Fang, Yaojun Liu, Yuqin Zhang, Huang Li

**Affiliations:** 1Affiliated Union Hospital of Fujian Medical University, Fuzhou 350122, China; 2Pharmacy College, Fujian University of Traditional Chinese Medicine, Fuzhou 350122, China

**Keywords:** *Pseudognaphalium affine*, COPD, pro-inflammatory, TNF-α, NF-κB pathway

## Abstract

Chronic obstructive pulmonary disease (COPD) is a chronic respiratory disease with limited therapeutic options. *Pseudognaphalium affine* (D. Don) Anderb. is a medicinal and edible plant used to treat cough, asthma, and COPD for a long time in folk medicine. The objective of this study is to evaluate the effect of *Pseudognaphalium affine* (D. Don) Anderb. extract (GAE) and investigate the possible underlying mechanism in vivo and in vitro. In vivo, the administration of GAE in a rat COPD model could significantly ameliorate lung damage and pulmonary function by inhibiting the production of pro-inflammatory cytokines. Western blot and real-time PCR results showed that GAE could suppress nuclear translocation of nuclear factor-kappa B (NF-κB), which indicated that GAE down-regulated the NF-κB pathway. Moreover, GAE protected against tumor necrosis factor (TNF)-α induced inflammatory response in BEAS-2B and inhibited the NF-κB pathway. All data suggested that GAE exhibited its anti-COPD effect by inhibiting pro-inflammatory cytokines, which may be associated with the inhibition of the NF-κB pathway.

## 1. Introduction

Chronic obstructive pulmonary disease (COPD) is a chronic respiratory disease associated with chronic pulmonary inflammation and irreversible airflow restriction [1]. COPD is a major cause of morbidity and mortality, which has become a significant economic and social burden in many countries around the world [1,2]. Dyspnea, chronic cough, chronic sputum production, wheezing, and chest tightness are the common symptoms of COPD. As a result of air pollution, active smoking, fumes, and chemicals from the industries. Furthermore, it has been established that outdoor air pollution is a significant risk factor for the occurrence and worsening of respiratory disorders. Pollutants generated by vehicles and factories account for most of the air pollution. According to studies, the impact of air pollution is more prevalent in women than in men. A cross-sectional cohort study found a link between traffic exposure and poor lung function, but only in women was it statistically significant [1,2]. The World Health Organization (WHO) predicts that this will become the third leading cause of death in the world in 2030, and there may be over 5.4 million deaths annually from it in 2060 due to the increasing numbers of smokers and aging populations [3,4]. COPD is a worldwide public health challenge. To date, the drugs for the treatment of COPD are mainly glucocorticoids and bronchodilators. Current strategies to prevent COPD progression or reversal are limited [5]. In the 21st century, COPD is becoming one of the major health problems. An increasing number of alternative methods and appropriate treatments are being promoted to COPD patients.

Traditional Chinese medicine (TCM) contains a large number of bioactive ingredients which has positive effects on the treatment of patients with COPD [6]. *Pseudognaphalium affine* (D.Don) Anderb., also known as *Gnaphalium affine* D. Don, a traditional Chinese herb medicine, is usually used to treat cough, asthma, rheumatic arthritis, and gout for a long time in folk medicine [7,8]. Additionally, *P. affine* contains antioxidant property and is used as a potential source of nutraceuticals in the food industry. The extract of *P. affine* also contains an anti-inflammatory property [7,8]. It is also known as cudweed or Ching Ming vegetable, as a medicinal and edible plant in China. Folk prescription and clinical practice showed *P.* affine is usually used as a supplementary treatment to respiratory diseases, such as cure cough and asthma [9,10,11]. *P. affine* contains abundant compounds, such as flavonoids, polyphenols, polysaccharides, essential oil compounds, terpenoids, and other chemical compounds [12,13,14]. Among them, some compounds were reported to possess anti-complementary activity [12,13], antimicrobial [7], anti-inflammatory [15,16,17,18], antioxidant [7,13,19], antitussive and expectorant activities [20], and expectorant and anti-xanthine oxidase activities [21]. Our previous studies also showed that the total flavonoids of *P. affine* can prolong the cough incubation period, reduce the number of coughs, and increase the excretion of phenol red in mice [15], and *P. affine* extract can improve airway inflammation in COPD rats [16]. However, the underlying mechanisms of *P. affine* for the treatment of COPD remain unclear and require further investigation. 

The precise pathogenesis of COPD is still unknown, but airway inflammation is an essential core of COPD formation and development. Therefore, in this study, we aim to investigate the therapeutic effect of the *P. affine* extract (GAE) on COPD rats and the anti-inflammatory mechanisms in airway epithelial cells induced by TNF-α.

## 2. Results

### 2.1. Effect of GAE on the Pulmonary Function and Pathological Changes of Lung Tissue

Pulmonary degeneration is the characteristic feature of COPD, so we evaluated the effects of GAE on lung tissue morphology and lung function. As shown in Table 1, compared with the control group, there was a significant decrease in pulmonary function (FEV0.3, FVC, and FEV0.3/FVC) of the model group (*p* < 0.01). In contrast, GAE treatment resulted in an increase in these parameters. As shown in Figure 1, H&E staining analysis of lung tissues demonstrated that exposure to smoke and LPS induced serious pathological changes in the model group compared to the control group, such as decrease in the alveolar number, alveolar destruction, alveolar cavity expansion, bronchi wall thickness, and bronchi stenosis. In comparison, lung damage was found to be markedly ameliorated in the GAE group.

### 2.2. Effect of GAE on the Levels of Cytokines in the Serum and BALF

COPD is associated with a series of inflammatory reactions; then, the levels of pro-inflammatory cytokines in the serum and BALF were measured to investigate the pulmonary inflammatory responses. As shown in Figure 2, the level of IL-1β, IL-6, IL-8, IL-17, TNF-α, and TGF-β in the serum and BALF of the model group rats were found to be significantly increased (*p* < 0.01), while these increases were obviously inhibited by the administration of GAE (*p* < 0.05 or *p* < 0.01). 

### 2.3. Effect of GAE on the Inflammation in BEAS-2B Cells Induced by TNF-α

To confirm that the protection afforded by GAE also contributes to anti-inflammation in TNF-α induced inflammatory response in BEAS-2B, the levels of cytokines were measured. As shown in Figure 3, IL-1β, IL-6, and IL-8 were induced following exposure of BEAS-2B to TNF-α. Interestingly, the mRNA levels of these cytokines were significantly down-regulated after GAE intervention. 

### 2.4. Effect of GAE on the NF-κB Pathway

Inflammation is an essential core of COPD formation and development. Herein, we focused on the NF-κB signaling pathway, which has been considered a vital role in inflammatory responses in COPD. As shown in Figure 4 following exposure to cigarette smoke and LPS, NF-κB mRNA levels were markedly increased compared with that of the control group, and it was restored to a normal level by GAE treatment. The result was verified by Western blotting assay (Figure 4). Next, we also evaluated the regulators of NF-κB in the cytosol, such as IκB, p-IκB, and IKKβ. As expected, IκBα expression decreased in COPD and increased in GAE treatment groups, while p-IκBα and IKKβ expression increased in COPD and decreased in GAE treatment groups (*p* < 0.05 or *p* < 0.01), respectively. Furthermore, we examined the effect of GAE on the NF-κB signaling pathway of TNF-α-treated BEAS-2B. Notably, GAE decreased the levels of p-IκBα, IKKβ, and NF-κB and increased IκBα expression compared with the TNF-α group (Figure 5). The results were consistent with in vivo results.

## 3. Discussion

COPD has become a global public health issue with complex pathogenesis, and up to now, there is still a lack of effective treatment. TCM exerted an irreplaceable role in the treatment of complex chronic diseases due to their abundant bioactive compounds and targeting of multiple pathways and multiple targets [22]. *P. affine*, as a medicinal and edible plant in China, has a long history of application in folk. Our previous studies also showed that the total flavonoids of *P. affine* can prolong the cough incubation period, reduce the number of coughs, and increase the excretion of phenol red in mice and *P. affine* extract can improve airway inflammation in COPD rats [15,16]. However, the mechanisms by which *P. affine* regulates its anti-COPD effects are not illuminated. In the present study, rats exposed to cigarette smoke and LPS were prepared as a COPD model as previous study [16] and were treated with GAE to investigate its anti-COPD effect. The results highlighted that GAE treatment can alleviate the symptoms of COPD rats by improving lung function and ameliorating pathological changes. In addition, we found that administering GAE can down-regulate classical pro-inflammatory cytokines. The results were consistent with a previous study [15,16]. COPD is associated with a series of inflammatory reactions [23]. In this study, we employed a TNF-α exposure method to mimic a harmful inflammatory stimulation in the airway epithelial cells [24]. To verify its anti-inflammation effect, in this study, we validated the anti-inflammatory and regulatory effects of GAE on TNF-α-induced inflammatory responses in airway epithelial cells. The results showed that GAE suppressed the inflammatory response by decreasing the levels of IL8, IL6, and TNF-α. The anti-COPD effect afforded by GAE also contributes to anti-inflammation in TNF-α induced inflammatory response in BEAS-2B. 

These results are consistent with the previous similar studies using in vivo and in vitro models of other diseases. In terms of in vivo studies, Huang et al. demonstrated that the level of pro-inflammatory cytokines, such as TNF-α, IL-1β, and COX-2, was obviously downregulated by *P. affine* treatment in collagen-induced arthritis (CIA) rats [25]. Zhang et al. reported that *P. affine* significantly decreased the expression of IL-1β and TNF-α in MSU Crystal-induced mice [8]. In vitro studies, Seong et al. found that *P. affine* reduced the expression of iNOS, COX-2, and MAPKs in LPS-stimulated RAW264.7 cells [17]. Ryu et al. reported that *P. affine* inhibited LPS-induced NO production in RAW264.7 macrophage cells [18]. Zhang et al. also reported that *P. affine* significantly decreased the levels of TNF-α, IL-1β, and COX-2 in LPS-stimulated NR8383 cells [8]. With the evidence above, we verified the anti-COPD effects of *P. affine*, which may be associated with the regulation of cytokines.

NF-κB is a multifunctional nuclear transcription factor, and many experiments have shown that NF-κB plays an important role in COPD [26]. NF-κB transcriptional activity is carefully inhibited by binding to IκB in an inactive condition, while costimulating molecules, such as LPS, TNF-α bind to receptors TLRs, TNFRs, and IL-1R, which can lead to the IKKβ complex phosphorylating the inhibitory IκB proteins. Then, NF-κB is dissociated from IκBα and is translocated into the nucleus. Subsequently, it induces transcriptional upregulation of various inflammatory mediators [27]. Downregulation of IκBα was the classic negative feedback of NF-κB activation, and the nuclear translocation of phosphorylated p65 was consistent with the IκBα phosphorylation. Drug therapies targeting the NF-κB activation pathway have attracted attention as an interesting therapy to treat lung disease that occurs due to an inflammatory response [28]. Several studies have demonstrated that natural compounds, herbal extracts, and herbal formulas could exhibit protective effects against COPD by regulating inflammatory responses via NF-κB signaling [24,29,30]. Our results show that GAE could reverse the up-regulation of the expression level of p-IκBα, IκK, and NF-κB in COPD rats and BEAS-2B. The results agreed with the changes in proteins associated with the NF-κB signaling pathway induced by LPS on RAW264.7 cells [17]. Our results have shown that to strengthen the evidence that the NF-κB signaling pathway may play a role in the effects of *P. affine*. Nevertheless, further verifications are needed in our future study to elucidate the relationship between the anti-COPD effect and the NF-κB signaling pathway of *P. affine*.

Plants of the *Gnaphalium* genus are an herb distributed worldwide with about 200 species. Nowadays, a large of chemical ingredients have been isolated from plants, and most share a similar chemical composition. They mainly include flavonoids, terpenoids (sesquiterpenes, diterpenes, and triterpenes), phytosterols, caffeic acid derivatives, and other compounds, and they also demonstrate a variety of pharmacological effects [31]. Liu et al. found that Dihydroquercetin can suppress cigarette-smoke-induced ferroptosis in the pathogenesis of COPD by activating the Nrf2-mediated pathway [32]. Lin et al. pointed out that ursolic acid effectively attenuated CS-induced mice emphysema, which may be beneficial by down-regulation of the PERK pathway and up-regulation of Nrf2 signaling [33,34]. In this study, 17 main compounds of flavonoids and acids were identified in GAE. They are the important material basis of GAE. This study uncovered that GAE could effectively protect against COPD by regulating inflammatory responses via inhibiting the NF-κB signaling pathway from another point of view to explain the therapeutic mechanism of the *Gnaphalium* genus.

## 4. Materials and Methods

### 4.1. Chemicals and Reagents

*P. affine* was purchased from Bozhou Great Northwest Pharmaceutical Co., Ltd. (Anhui, China). Lipopolysaccharide (LPS) was purchased from Sigma (St. Louis, MO, USA). Cigarettes were obtained from Huangshan Tobacco Industry (Hongqi Canal^®^ Filter tip cigarette; tobacco type, tar: 10 mg; nicotine content: 1.0 mg; carbon monoxide: 12 mg; Anhui, China). Dexamethasone acetate tablets were obtained from Zhejiang Xianju Pharmaceutical Co. Ltd. (Zhejiang, China). Interleukin (IL)-1β, IL-6, IL-8, IL-17, tumor necrosis factor (TNF)-α, and transforming growth factor (TGF)-β enzyme-linked immunosorbent assay (ELISA) kits were purchased from Boster Biological Engineering (Wuhan, China). IκB kinase (IKK) β antibodies were purchased from Abcam (Waltham, MA, USA), nuclear factor-kappa B (NF-κB), IκBα and p-IκBα antibodies were purchased from Cell Signaling Technology (Danvers, MA, USA). Anti-Rabbit/Mouse IgG secondary antibodies were purchased from LuLong Biotech Co. (Xiamen, China). RevertAid First Strand cDNA Synthesis Kit was purchased from Thermo Fisher Scientific (Shanghai, China).

### 4.2. Animals

Sprague-Dawley rats (200 ± 20 g) were purchased from Shanghai Jihui experimental animal breeding Co., Ltd. [scxk (Shanghai) 2017-0012], and all rats were housed in the animal experiment center of Fujian University of traditional Chinese medicine. The animal experiments were conducted with the approval of the Experimental Animal Care and Ethics Committee of Fujian University of traditional Chinese medicine (DWLL20170012010982).

### 4.3. Pseudognaphalium affine Extract (GAE) Preparation and Analysis

The extraction and analysis method has been slightly modified by the previous reported [8]. The dried *P. affine* (whole plants) was cut into slices before use. Subsequently, *P. affine* (50 g) was extracted with 400 mL, 75% ethanol (1:8, *w*/*v*) 2 times, 1 h for each time. The extract was filtered and combined, then the filtrate was concentrated by vacuum rotary evaporation and totally dried by vacuum freeze drying. GAE was determined by ultra-performance liquid chromatography (UPLC), and 17 chemical constituents of GAE were identified according to the spectrograms and retention times of their standard substances (Figure 6).

### 4.4. Establishment of COPD Model and Drug Treatment

The rat model of COPD was established as previously described [16]. Rats were randomly divided into the control group, model group, GAE group, and dexamethasone group (positive control drug), with 9 rats in each group. Except for control group, the rats in other groups were placed in a chamber and exposed to the smoke of 5 cigarettes for 30 min twice daily, with 3 h of smoke-free intervals during the 4 weeks. Then, 0.2 mL LPS (1 mg/mL) was inoculated on the 1st and 15th day of the experiment through the trachea. Control rats received the same treatment but were exposed to air and normal saline rather than cigarette smoke and LPS. Rats were orally administered normal saline, GAE (75, 150, and 300 mg/kg), or dexamethasone (27 mg/kg) every day from the second week. In the fourth week, rats were sacrificed to collect lung tissue, blood, and bronchoalveolar lavage fluid (BALF) after anesthesia.

### 4.5. Pulmonary Function Analysis

To assess pulmonary function, 0.3 s forced expiratory volume (FEV0.3), and 0.3 s forced expiratory volume to forced vital capacity (FEV0.3/FVC) were detected using computer-controlled unrestrained pulmonary function-testing plethysmography.

### 4.6. Histopathology

The left lower lobe was collected and fixed in 4% paraformaldehyde neutral buffer solution, then embedded in paraffin, and finally sliced into 4 μm sections. Sections were stained with Mayer’s hematoxylin and 1% eosin alcohol solution (H&E staining), and the pathological changes of lung tissue were observed under a light microscope.

### 4.7. Cytokine ELISA Assay

The level of IL-1β, IL-6, IL-8, IL-17, TNF-α, and TGF-β in the serum and BALF supernatants were measured with ELISA kits according to the manufacturer’s instructions.

### 4.8. Cell Culture and Treatment

The human bronchial cell line BEAS-2B (Shanghai Fuheng Biotechnology Co., Ltd., Shanghai, China) was cultured in DMEM with fetal bovine serum (10%) at 5% CO_2_ and 37 °C. The cells were treated with GAE (50, 100, and 200 μg/mL) or vehicle (cell culture/0.1% dimethyl sulfoxide) 24 h prior to stimulation with TNF-α (10 ng/mL). Samples for analysis by real-time PCR and Western blotting were collected at 24 h.

### 4.9. Real-Time PCR Assay

Total RNA was extracted from 100 mg lung or cultured cells using Trizol. Then, RNA was reversed and transcribed into cDNA using RevertAid First Strand cDNA Synthesis Kit. cDNA was used as a template for Real-rime PCR amplification, and it was then carried out on ABI 7900HT Real-Time PCR System (Applied Biosystems Inc., Foster City, CA, USA) based on general fluorescence detection by SYBR Green. The primer sequences of the target gene are shown in Table 2. The relative transcriptional level of the target gene was calculated using the 2^−ΔΔCt^ method and normalized to the GAPDH gene.

### 4.10. Western Blotting Assay

Proteins were extracted from lung or cultured cells lysed with RIPA lysis buffer on ice. Subsequently, protein samples of equal quality were separated using 10% SDS-PAGE gel and electro-transferred to PVDF membranes. Membranes with proteins were blocked with 5% nonfat milk and then incubated with primary and secondary antibodies. Ultimately, signals were visualized using the ECL Western blotting detection reagents in Bio-Rad Imaging System. The relative expression level of the target gene was calculated and normalized to β-actin.

### 4.11. Statistical Analysis

All the data were analyzed using SPSS software (25.0) and expressed as the mean ± standard deviation. Comparisons among multiple groups were performed by a one-way analysis of variance, followed by a post hoc Tukey’s test. *p* values < 0.05 were considered statistically significant differences.

## 5. Conclusions

In summary, these results demonstrated that GAE exhibit a good potent anti-COPD effect by regulating inflammatory responses. The underlying mechanism of GAE may be related to the NF-κB signaling pathway.

## Figures and Tables

**Figure 1 molecules-27-08243-f001:**
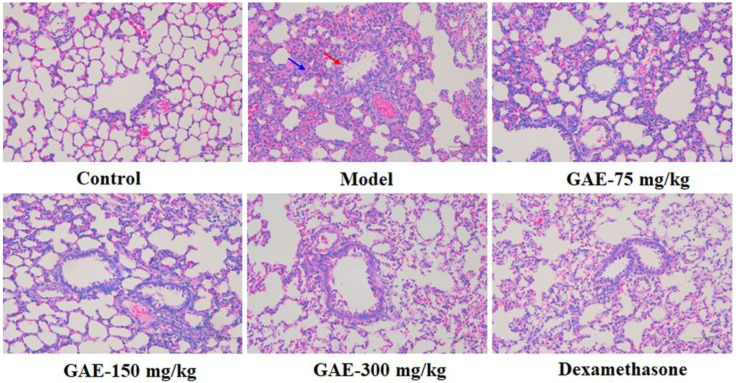
Effect of GAE on the pathological changes in COPD rats. Representative images of lung tissues’ morphology stained by H&E. Original magnification × 100. Bronchi wall thickness and bronchi stenosis (red arrows) and alveolar destruction (blue arrow) were observed.

**Figure 2 molecules-27-08243-f002:**
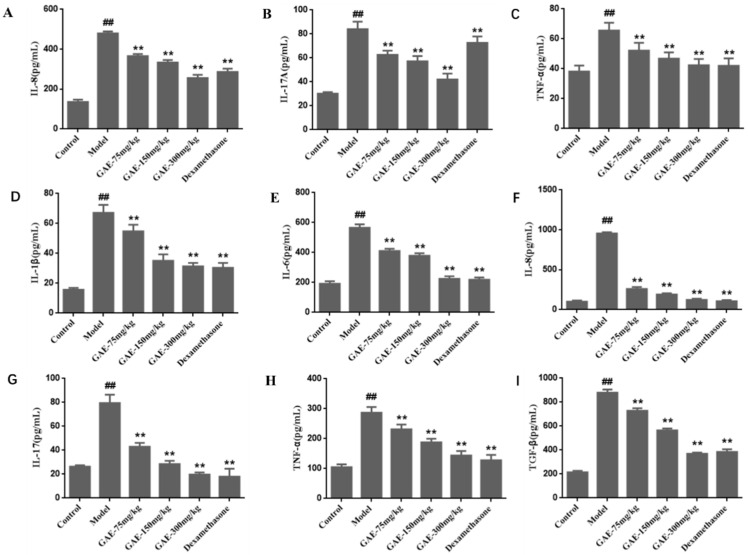
Effect of GAE on the levels of cytokines in the serum and BALF. The levels of IL-1β, IL-6, IL-8, IL-17, TNF-α, and TGF-β in the serum (**D**–**I**) and BALF (**A**–**C**) of rats were detected by ELISA according to the manufacturer’s instructions. Data were presented as the mean ± standard deviation, n = 6. vs. control, ^##^
*p* < 0.01, vs. model, ** *p* < 0.01.

**Figure 3 molecules-27-08243-f003:**
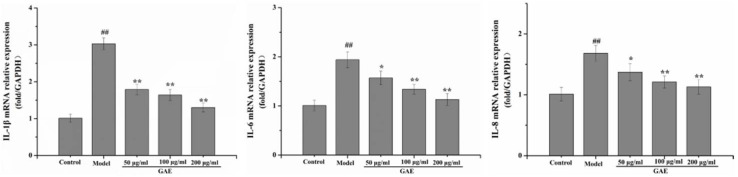
Effect of GAE on the inflammation in BEAS-2B cells induced by TNF-α. The mRNA levels of IL-1β, IL-6, and IL-8 in cells were detected by real-time PCR. Data were presented as the mean ± standard deviation, n = 3. vs. control, ^##^
*p* < 0.01, vs. model, * *p* < 0.05, ** *p* < 0.01.

**Figure 4 molecules-27-08243-f004:**
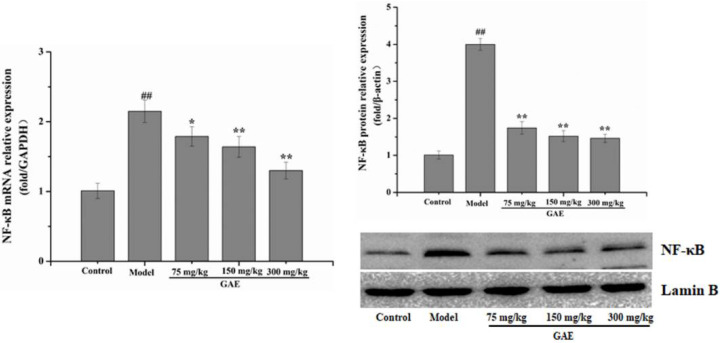
Effect of GAE on the NF-κB Pathway in the lung tissue of COPD rats. The levels of NF-κB, IκB, p-IκB, and IKKβ were detected. Images are representative of n = 3 per group. Data were presented as the mean ± standard deviation. vs. control, ^##^
*p* < 0.01, vs. model, * *p* < 0.05, ** *p* < 0.01.

**Figure 5 molecules-27-08243-f005:**
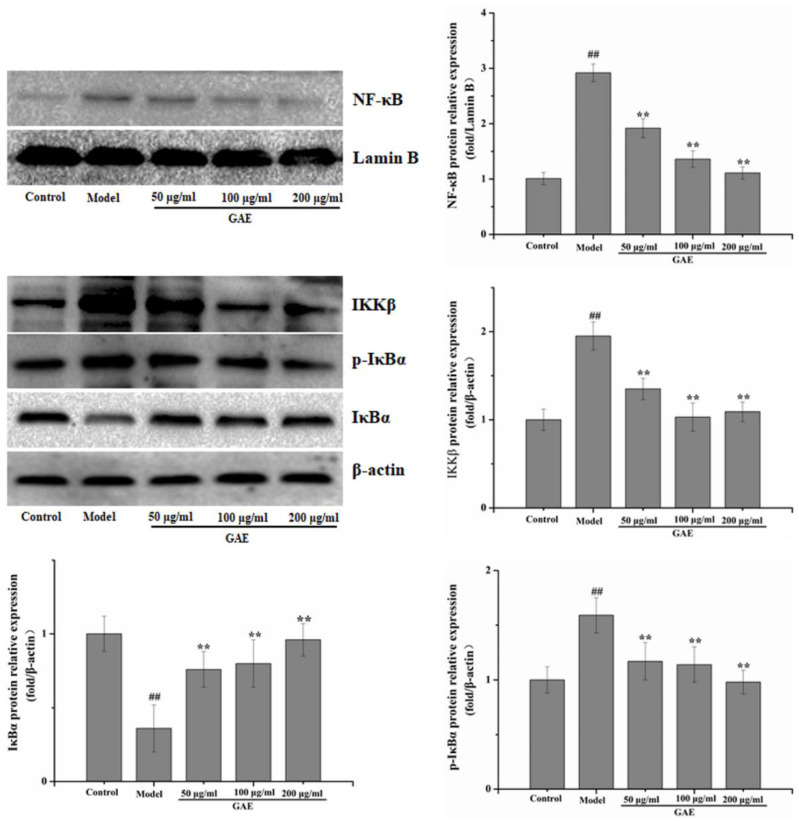
Effect of GAE on the NF-κB Pathway in BEAS-2B cells induced by TNF-α. The levels of NF-κB, IκB, p-IκB, and IKKβ were detected by Western blotting. Data were presented as the mean ± standard deviation. Images are representative of n = 3 per group. vs. control ^##^
*p* < 0.01, vs. model, ** *p* < 0.01.

**Figure 6 molecules-27-08243-f006:**
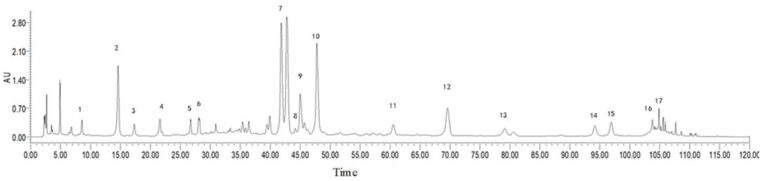
UPLC analysis of GAE. Peak labeling represents 17 compounds identified. They were protocatechuic acid (1), chlorogenic acid (2), caffeic acid (3), 1,3-O-dicaffeoyl quinic acid (4), isoflavin glycosides (5), isozephyrin (6), 1,5-O-dicaffeoyl quinic acid (7), apigenin-7-O-β-D-glucopyranoside (8),quercetin-4’-O-β-D-glucopyranoside (9), luteolin-4’-O-β-D-glucopyranoside (10), luteolin (11), quercetin (12), 1,4,5-O-tricaffeoylquinic acid (13), kaempferol (14), 5,7-dihydroxy-3,8-dimethoxyflavone (15), apigenin (16), 3,8,4’-trimethoxy-5,7 dihydroxyflavone (17).

**Table 1 molecules-27-08243-t001:** Effect of GAE on the pulmonary function in COPD rats ($\bar{X}$ ± s).

Groups	FEV0.3 (mL)	FVC (mL)	FEV0.3/FVC (%)
Control	7.26 ± 0.47	9.02 ± 0.23	85.44 ± 5.11
Model	3.73 ± 0.35 ^##^	5.24 ± 0.11 ^##^	51.58 ± 3.18 ^##^
GAE-75 mg/kg	4.34 ± 0.18 *	6.25 ± 0.62	54.14 ± 2.66
GAE-150 mg/kg	5.60 ± 0.33 **	6.61 ± 0.23 **	68.62 ± 5.19 *
GAE-300 mg/kg	5.69 ± 0.22 **	7.30 ± 0.41 **	74.53 ± 4.35 *
Dexamethasone	5.58 ± 0.24 **	7.28 ± 0.67 **	70.34 ± 2.48 *

Notes: vs. control, ^##^
*p* < 0.01, vs. model, * *p* < 0.05, ** *p* < 0.01.

**Table 2 molecules-27-08243-t002:** Primer sequences of mRNA.

Gene	Primer	Primer Sequence
GAPDH	Forward primer	5’-AGCCCAGAACATCATCCCTG-3’
	Reverse primer	5’-CACCACCTTCTTGATGTCATC-3’
NF-κB	Forward primer	5’-CATCAAGCGTACGTGCGTA-3’
	Reverse primer	5’-CTGATGCGTCTGAGATCTA-3’
IL-1β	Forward primer	5’-ATGACCTGTTCTTTGAGGCTGAC-3’
	Reverse primer	5’-CGAGATGCTGCTGTGAGATTTG-3’
IL-6	Forward primer	5’-GACCAAGACCATCCAACTCATC-3’
	Reverse primer	5’-ACATTCATATTGCCAGTTCTTCGTA-3’
IL-8	Forward primer	5’-GACTGTTGTGGCCCTGGAG-3’
	Reverse primer	5’-CCGTCAAGCTCTGGATGTTCT-3’
TNF-α	Forward primer	5’-CAGGCGGTGCCTATGTCTC-3’
	Reverse primer	5’-CGATCACCCCGAAGTTCAGTAG-3’

## Data Availability

The data presented in this study may be available on reasonable request from the corresponding author.

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
