# Peer review of "Pseudognaphalium affine* Extract Alleviates COPD by Inhibiting the Inflammatory Response via Downregulation of NF-κB"

_molecules, 2022, doi:10.3390/molecules27238243_

Round 1

Reviewer 1 Report

Nowadays, research on COPD has increased substantially, seeking less aggressive medications. In the case of the Gnaphalium genus, the chemical compounds it presents and their pharmacological characteristics are known (for example Sheng, 2013), in addition, the molecular events involved in the pathogenesis of COPD and their modulation by plant products are known. Many plants have been investigated in relation to this condition, extracting chemical compounds and detailing their mechanisms of action. Therefore, this article needs to be substantially improved by making comparative analyzes between what the authors find with what is reported in other plant species studied.

It is necessary to check if the name of the genus is correct, because according to the Tropicos page it should be Pseudognaphalium.

Author Response

I appreciate for your careful reviews and constructive comments for improving our manuscript. Indeed, I have seriously read all the comments. All the comments are constructive and valuable. I have learned much from these valuable comments. Irespected all the comments and attempted to make revisions to the best of our ability. Here, I provided a point-by-point response to the comments in the revised version.

Q1. Nowadays, research on COPD has increased substantially, seeking less aggressive medications. In the case of the Gnaphalium genus, the chemical compounds it presents and their pharmacological characteristics are known (for example Sheng, 2013), in addition, the molecular events involved in the pathogenesis of COPD and their modulation by plant products are known. Many plants have been investigated in relation to this condition, extracting chemical compounds and detailing their mechanisms of action. Therefore, this article needs to be substantially improved by making comparative analyzes between what the authors find with what is reported in other plant species studied.

Response: Thanks for your valuable comments. We added the discussion about this in the discussion section of the manuscript.

Q2. It is necessary to check if the name of the genus is correct, because according to the Tropicos page it should be Pseudognaphalium.

Response: Thanks for your valuable comments. Gnaphalium affine D.Don is the name used in Chinese Pharmacopoeia. I check it in the Plants of the World Online (https://powo.science.kew.org/), where show that Gnaphalium affine D.Don is a synonym of Pseudognaphalium affine (D.Don) Anderb., and Pseudognaphalium affine (D.Don) Anderb. is accepted names in the world. So, I have revised the name in the manuscript.

Reviewer 2 Report

Dear authors,

I am very satisfied about the outcome of this Ms, and I have only few comments.

Abstract:

The abstract can be improved by starting with the brief background of chronic obstructive pulmonary disease (COPD), followed by the aim of the study.

Figure 1. Effect of GAE on the pathological changes in COPD rats (HE is staining, magnification, × 100):

The histology figures should be presented with arrows, and the figure caption should be self-explanatory.

Discussion:

Line 170: p-IκB, could you please add α (p-IκBα)

It is preferable to extend the discussion regard to NF-κB and its regulators IκBα, p-IκB, and IκK.

4.11. Statistical analysis:

It is written that the data was expressed as the mean ± standard deviation, but in the figure caption it is written mean ± SEM! Could you please correct that.

Author Response

I appreciate  for your careful reviews and constructive comments for improving our manuscript. Indeed, I have seriously read all the comments. All the comments are constructive and valuable. I have learned much from these valuable comments. I respected all the comments and attempted to make revisions to the best of our ability. Here, I provided a point-by-point response to the comments in the revised version. 

Q1. Abstract: The abstract can be improved by starting with the brief background of chronic obstructive pulmonary disease (COPD), followed by the aim of the study.

Response: Thanks for your valuable comments. I have revised the abstract in the manuscript.

Q2. Figure 1. Effect of GAE on the pathological changes in COPD rats (HE is staining, magnification, × 100): The histology figures should be presented with arrows, and the figure caption should be self-explanatory.

Response: Thanks for your valuable comments. I have revised the figure caption of Figure 1, and re-upload the new histology figures with arrows.

Q3. Discussion: Line 170: p-IκB, could you please add α (p-IκBα). It is preferable to extend the discussion regard to NF-κB and its regulators IκBα, p-IκB, and IκK.

Response: Thanks for your valuable comments. I have revised the discussion about these, and added α to p-IκB.

Q4. 4.11. Statistical analysis: It is written that the data was expressed as the mean ± standard deviation, but in the figure caption it is written mean ± SEM! Could you please correct that.

Response: Thanks for your valuable comments. I have revised the mistake in the figure caption.

Round 2

Reviewer 1 Report

The authors improved their manuscript and it can be published.